# Occurrence and Dietary Risk Assessment of Mycotoxins in Most Consumed Foods in Cameroon: Exploring Current Data to Understand Futures Challenges

**DOI:** 10.3390/foods12081713

**Published:** 2023-04-20

**Authors:** Isabelle Sandrine Bouelet Ntsama, Chiara Frazzoli, Guy Bertrand Pouokam, Vittorio Colizzi

**Affiliations:** 1Advanced Teacher’s Training College for Technical Education, University of Douala, Douala P.O. Box 1872, Cameroon; 2Laboratory of Food Science and Metabolism, Department of Biochemistry, Faculty of Sciences, University of Yaoundé I, Yaoundé P.O. Box 812, Cameroon; 3Department of Cardiovascular and Endocrine-Metabolic Diseases, and Ageing, Istituto Superiore di Sanità, 00161 Rome, Italy; 4Department of Biochemistry, Faculty of Medicine and Biomedical Sciences, University of Yaoundé I, Yaoundé P.O. Box 1364, Cameroon; 5Nutrition and Food Safety and Wholesomness (Noodles Cameroon), Yaoundé P.O. Box 3746, Cameroon; 6Interdipartimental Centre for Comparative Medicine, University of Rome Tor Vergata, 00133 Rome, Italy

**Keywords:** mycotoxins, exposure, total diet studies, risk assessment, primary prevention, foods

## Abstract

Mycotoxins are naturally occurring toxins that contaminate different crops and foodstuffs under certain circumstances during harvesting, handling, storage, and processing. Neither the dietary intake of mycotoxins in Cameroon is well characterized, nor its health effects on the consumers. This review is intended to be the first milestone towards national risk management of mycotoxins. It is noteworthy that mycotoxins contaminate the main staple foods of Cameroonian communities, which are also often used as complementary foods for infants, young children, and people with compromised immune systems (e.g., HIV/AIDS), thus calling for urgent intervention in primary and secondary prevention. Very few data exist on mycotoxin contamination in Cameroonian agricultural commodities and food items. Only 25 studies from 14 different authors have been published in the last decade. On the basis of available data in Cameroon, the Estimated Daily Intake (EDI) of major mycotoxins in foods for Aflatoxins was 0.0018–14.2 µg/kgbw/day in maize, 0.027–2.36 µg/kgbw/day in cassava, and 0.023–0.1 µg/kgbw/day in groundnuts. The estimated daily intake of fumonisins was 0.12–60.6 µg/kgbw/day in maize and 0.056–0.82 µg/kgbw/day in beans. Based on the estimated distribution of human exposure levels by food, maize and cassava are the major sources of exposure and should be prioritized, followed by beans and spices. This estimate will be updated along with improvements on the national database on mycotoxin contamination of Cameroonian foods.

## 1. Introduction

Mycotoxins are naturally occurring toxins produced by certain molds (fungi, mainly *Aspergillus, Penicillium*, and *Fusarium*) that grow on a variety of different crops and foodstuffs (e.g., maize, peanut/groundnut, sorghum, millet, wheat, and rice, but also spices, dried fruits, apples, coffee beans, and cocoa) under certain circumstances (e.g., warm and humid conditions) during harvesting, handling, storage, and processing. According to [1], some mycotoxins such as Deoxynivalenol (DON), Zearalenone (ZEN), and Fumonisin (FB) may start to contaminate grains at the field/or preharvest while Aflatoxin (AF) and Ochratoxin (OT) can also occur during storage due to improper postharvest handling.

Exposure to mycotoxins occurs mostly by ingestion and poses a serious health threat to both humans and livestock, thus representing a main health issue [2,3]. Among mycotoxins, the ones of major agroeconomic (food safety, food security) and public health importance are Aflatoxins (AFB1, AFB2, AFG1, AFG2, AFM1, and AFM2), Zearalenone (ZEN), Fumonisin B1 (FB1), Patulin, Ochratoxin A (OTA), and Deoxynivalenol (DON) [4,5]. The adverse health effects range from acute poisoning to disorder of the central nervous, cardiovascular, and pulmonary systems and intestinal tract to death [6] and long-term effects, such as immune deficiency and cancer [7].

AFs are hepatotoxic and immunosuppressive, and aflatoxicosis is a disease caused by AFs leading to jaundice and, in severe cases, death. Repetitive incidents have occurred in Kenya (during 1981, 2001, 2004, and 2005), India, and Malaysia [8,9]. AFB_1_ has been extensively linked to human primary liver cancer in which it acts synergistically with Hepatitis B Virus infection and was classified by the International Agency for Research on Cancer (IARC) as a human carcinogen (Group 1 carcinogen) [10]. This combination represents a heavy cancer burden in developing countries. OTA is carcinogenic, genotoxic, nephrotoxic, and immunosuppressive. FBs are carcinogenic, hepatotoxic, nephrotoxic, and immunosuppressive. DON causes vomiting, nausea, diarrhea, and reproductive effects.

Zearalenone is a mycotoxin with immunotoxic, hepatotoxic [11], and estrogenic effects [12]. The activity of ZEN in living organisms depends on the immune status of the organism and the state of the reproductive system (adolescence or pregnancy stage) [13].

Patulin causes neurologic and gastrointestinal perturbations. The effect of mycotoxins on health depends not only on the magnitude/dose and frequency of exposure but also on age, sex, weight, nutritional status, exposure to infectious agents, and pharmacologically active substances [6]. At lower concentrations, the effects of mycotoxins are more protean. They reduce the growth rate of young animals, and some interfere with native mechanisms of resistance and impair immunologic responsiveness, making the animals more susceptible to infection [5]. They are considered to have immunosuppressive effects and inhibit DNA synthesis [14].

Citrinin (CIT) is a severe problem in countries with a hot and humid climate, as it is a major source of food poisoning after fungal contamination. Citrinin has been known to be nephrotoxic, hepatotoxic, and carcinogenic to humans and animals. After ingestion of CTN-contaminated food, it affects the kidneys, targeting the mitochondrial respiratory chain [15]. Citrinin was associated with yellow rice syndrome in Japan in 1971 because of the regular presence of *P. citrinum* in this food product [5,15]. It has also been considered responsible for nephropathy in pigs and other animals, although its acute toxicity varies depending on the animal species.

Penicillic acid has been found in large quantities in high-moisture corn stored at low temperatures. Penicillic acid has low oral toxicity. The concern about this toxin in foods is related to its structural similarity to known carcinogens, such as patulin, and its carcinogenic effect on rats when injected subcutaneously. The potencies of penicillic acid and patulin as carcinogens, however, are much lower than aflatoxins [14,16].

Preservation of food commodities in Cameroon is still of great concern in many communities, as recently highlighted by the narrative prevention project of the NGO NOODLES [16]. Food agricultural and manufacturing practices currently adopted in Cameroon are often very suitable to mold proliferation and mycotoxins production. In Cameroon, few studies investigated the contamination of foodstuffs by mycotoxins. A large number of commodities including cereals (such as maize, sorghum, and rice) and derivatives [17,18], such as cassava, peanuts, beans, legumes, spices, and farmed fish, have been found to be highly contaminated with mycotoxins [17,19,20]. Noticeably, when samples have low levels of individual mycotoxins, co-occurrence of mycotoxins was found to be of concern due to additive or synergistic effect, especially for commodities that are consumed on daily basis [17]. The presence of AFs and OTs has been reported in export products, such as coffee and cocoa [21,22,23].

Based on these preliminary data, this paper aims at providing an overview of the occurrence of mycotoxins in food commodities in Cameroon, as well as a picture of the human dietary exposure in Cameroon, and possible risk scenarios derived from not adopting good practices.

## 2. Characterization of the Country’s Climate, Agroecological Zones, and Production System

Cameroon is a country located in the sub-equatorial region of Africa. Cameroon shares a boundary in the north with Chad, in the east with the Central African Republic, in the south with Gabon, and in the west with Nigeria (Figure 1). Cameroon lies between longitude 6°00′ N and 16° N and latitude 1.65° E and 12°00′ E. Cameroon is therefore located 6°0′0″ N and 12°0′0″ E. Because Cameroon exhibits all the major climates and vegetation of the continent as mountains, desert, rain forest, savannah grassland, and ocean coastland, it is sometimes described as “Africa in miniature”. Cameroon can be divided into five agroecological zones (Figure 1) distinguishable by dominant physical, climatic, and vegetative features. The climate varies with terrain, from tropical along the coast to semi-arid and hot in the north. The coastal belt is hot and humid; it includes some of the wettest places on earth, such as Debundscha, located at the base of Mt. Cameroon, which has an average annual rainfall of about 10,287 mm. Agriculture is the backbone of Cameroon’s economy, employing 70% of its workforce and providing 44% of its gross domestic product and 30% of its export revenue. Cameroon produces several agricultural commodities for export and domestic consumption. The most important of these, which vary by agroecological zone (see Table 1), are cocoa, coffee, cotton, banana, rubber, palm oil, sugarcane, tobacco, tea, pineapple and peanuts for cash crops, and plantains, cassava, corn, millet, sorghum, yams, potatoes, sweet potatoes, dry beans, and rice for food crops. Animal husbandry is practiced throughout the country and is particularly important in the northern region. Estimates of milk production by FAO relate to net milk production from cows, sheep, goats, etc. [24]. Cattle constitute the primary source of milk and meat for the population. Production depends largely on the traditional pastoralists who rely almost exclusively on native pastures to meet the needs of their animals.

## 3. Methods

### 3.1. Eligibility Criteria, Data Sources, and Searching Strategies

By using an in-house developed knowledge management system that categorizes papers by assigning keywords. Studies were included based on the following criteria: (i) studies that reported food contamination by mycotoxins in Cameroon, (ii) studies that reported risk assessment related to mycotoxins, and (iii) articles that reported mycotoxins’ health effects in animals and humans. About 100 documents were selected for further reading as they have been categorized as food-related, out of which around 56 papers contained data that could be used in this review. The studies were searched using electronic databases, such as PubMed, Science Direct, web of science, and Google Scholar, which help to cover almost all published articles in related fields. Other data sources such as Hinari and Google were used to search for studies. The following keywords were used: mycotoxins, exposure, foods, dietary exposure, risk assessment, total diet studies, estimated daily intake, health effects of mycotoxins, and Cameroon. In this electronic database search, Boolean terms (AND/OR) were used to separate the keywords.

### 3.2. Consumption Data and Dietary Exposure

According to some data [24,25], approximately 25% of the world’s agricultural products is contaminated with mycotoxins. International standards and codes of practice to limit exposure to mycotoxins from certain foods are established by the Codex Alimentarius Commission based on the assessments of the WHO/FAO scientific expert committee (JECFA). Given that diet is the main source of exposure, dietary risk assessment is crucial to understand the extent of population exposure through eating habits. In addition, a dietary risk assessment will allow to assess the contribution of each food category and food items in the cumulate exposure of an individual, compared to Tolerable Daily Intake (TDI). To obtain a distribution of human exposure levels by food, we used two sources of Cameroonian food consumption data: (i) the Cameroonian Food Balance Sheet (FBS) from the FAO database (2018) [26] and (ii) data from the Total Diet Studies (TDSs) carried out in Cameroon (2008) [27].

#### 3.2.1. Consumption Data from Food Balance Sheet (2018)

Food Balance Sheet presents a comprehensive picture of the pattern of a country’s food supply during a specific reference period. The food balance sheet shows for each food item (i.e., each primary commodity and a number of processed commodities potentially available for human consumption) the sources of supply and its utilization. In particular, the quantity of foodstuffs produced in a country added to the quantity of food that is imported and adjusted to any change in stocks that may have occurred since the beginning of the reference period gives the supply available during that period in the country. Thereafter a distinction is made between (i) the food quantities that are exported or fed to livestock, used for seed, or put to manufacture for food use and nonfood uses, (ii) losses during storage and transportation, and (iii) food supplies available for human consumption. The per capita supply of each food item available for human consumption is obtained by dividing the food supply available for human consumption by the population actually consuming it. Data on per capita food supplies are expressed in terms of quantity and by applying appropriate food composition factors for all primary and processed products; they are also provided in terms of caloric value and protein and fat content [28].

#### 3.2.2. Consumption Data from Total Diet Studies (2008)

Total Diet Studies are used to assess long-term/chronic exposure to chemicals in the diet. Its primary purpose is to measure the average amount of a wide range of chemicals present in the diet that are ingested by various age/sex groups living in a country [29]. In the TDS implemented in Cameroon by Gimou et al. [28], the consumption data were obtained from the database of the Cameroonian Household Budget Survey. In the Household Budget Survey, household food consumption data were evaluated from food expenditures, i.e., all expenses during a given period were recorded (not the amount of food consumed) in a representative sample of Cameroonian households. The amount of food as purchased per adult equivalent in the households was then derived for an individual [28].

### 3.3. Estimated Dietary Intake

Dietary exposure to mycotoxins in food commodities was calculated through EDI (Estimated Dietary Intake) as described below: EDI (µg/kgbw/day) = mean conc. (µg/kg) × Food consumed (kg/day). Assuming 60 kg as the average body weight (bw), the daily consumption per kg of bw was calculated. The health risk characterization of each mycotoxin was performed by dividing the EDI by the TDI (µg/kgbw/day) of the corresponding mycotoxin (when available), as indicated in the equation: %TDI = (EDI/TDI) × 100.

## 4. Mycotoxins in Food Commodities in Cameroon

### 4.1. Cereals and Derived Products

Cereals are plants used extensively in food and feed manufacturing as a source of proteins, carbohydrates, and oils. Cereals and derived products are frequently contaminated by mycotoxins. The occurrence of mycotoxins in Cameroonian cereals, such as maize, sorghum, and rice, has been reported by many authors (Table 2). Among the most cultivated food crops, maize ranks third in terms of volume of production after cassava and banana. Maize is the staple food in the western highlands of Cameroon. Inappropriate postharvest practices increase the chances of harvested grains becoming infected with *Aspergillus* and *Fusarium* sp. [30], especially because most grains are harvested during the rainy season, which creates favorable conditions for infections.

All the studies [17,31,32,33,34] included in this review reported the presence of AFs (AFB1), FBs (FB1), Deoxynivalenol (DON), Zearalenone (ZEN), Patulin, and Total citrinin in maize and maize-derived products directly consumed by people (beer, porridge, and fufu). AF and FB contents in maize-based dishes (beer, porridge, and fufu) were dependent on the production process: processes including a sieving step led to lower mycotoxin concentrations [35], which in some cases exceeded the standard limit fixed by the European Commission for such products [36]. FB1, DON, and ZEN at levels up to 26.0, 1300, and 1100 µg/kg, respectively, were reported [18,19]. FB1 was showing the highest prevalence with a concentration increasing with storage time. AFs were observed with a high percentage of positive samples, with the level depending on the agroecological zone (AEZ).

The occurrence of mycotoxins in sorghum, a crop mainly produced and consumed in North Cameroon, and its derived products has been studied during the last decade [37]. OTA, DON, and FB1 were not found in the grains collected in North Cameroon by Djoulde [18], while AFB1 was detected in sorghum cultivated in the rainy season (0–230 µg/kg). In some locally produced artisanal sorghum by-products (beer, flour, baby’s beverage, and cake), mycotoxins were detected at levels ranging 0–250 µg/kg for AFB1, 0–45 µg/kg for OTA, and 0–538 µg/kg for DON, while, in a previous work, local sorghum beers were contaminated by both DON (0–730 µg/L) and FB1 (0–340 µg/L).

**Table 2 foods-12-01713-t002:** Mycotoxins in Cameroonian food and their derivatives.

	Region (AEZ)	Mycotoxin	Mycotoxin (μg/kg) Mean (min–max)	No. Positive Samples	References	Maximum Level in Food	Hazard Index (QLmax/ML)
Maize grains	West (AEZ 3) North West (AEZ 3) Center (AEZ 5)	FB1 DON ZEN	(300−26000) (<100−1300) (<50−110)	18	[19]	ZEN: 60 µg/kg to 75 µg/kg ** ZEN: 100 µg/kg EU ZEN: no Codex standard FB1: 4000 µg/kg ≠ FB1: 1000 µg/kg ** (Krska et al., 2019) AFB1: 2000 µg/kg (Codex) AFB1: 10 µg/kg (CX/CF 19/13/15 March 2019) AFB1: 5 µg/kg EU AFs: 20 µg/kg ** US FDA AFs: 4 µg/kg EU DON: 2000 µg/kg≠ DON: 750 µg/kg ** DON (cereals-based foods for children): 200 µg/kg DON (wheat, maize): 1000 µg/kg CIT: 2000 µg/kg EU	FB1: 7 to 26 (ML) DON 1 to 6 (children) ZEN: 2
	West (AEZ 3) North West (AEZ 3) Center (AEZ 5)	FBs DON ZEN	(50–26.000) (100–1300) (50–180)	18	[38]		DON 1 to 6 (children) ZEN: 3
	North West (AEZ 3) Center (AEZ 5) Littoral (AEZ 4)	FB1 ZEN DON AFs	3684 (37–24225) 69 (28–273) 59 (18–273)	26 31 29 22	[17]		FB1: 6 to 24 (ML) ZEN: 4 DON (children): 1
	North West (AEZ 3)	FB1	508 (2–2313)	37	[32]		FB1(ML): 0.5 to 2
	West (AEZ 3) North West (AEZ 3) Center (AEZ 5) North (AEZ 1)	AFs	1 (≤2–42)	6	[33]		AFs: 4.2
	Littoral (AEZ 4) North (AEZ 1)	Total Citrinin	(5.7–6.5) (2.2–3.0)	NI	[34]		CIT: 0.0032 ZEN: 0.97 to 1.61
	Littoral (AEZ 4) North (AEZ 1)	ZEN	(7.6–97.0)	NI	[34]		
	Littoral (AEZ 4) North (AEZ 1)	FBs	(64.4–71.6) (19.0–27.1)	NI	[34]		
Maize kernels	North (AEZ 1) West (AEZ 3)	FBs AFB1 ZEN DON	(10–5412) (6–645) (27–334) (27–3842)	66 36 23 19	[31]	-	FBs: 2.7 to 5.4 AFB1:0.32 to 129 ZEN: 3.34 to 5.56 DON: 1.9 to 3.8
Maize-based dishes	Center (AEZ 5)	AFs FBs	(0.8–20) (10–5990)	22 22	[39]	AFB1 (infants and young children): 0.10 µg/kg ** DON: 750 µg/kg **	AFB1: 200 FBs: 1.49 to 5.99
Maize-*fufu*	West (AEZ 3)	AFB1 DON FB1 NIV PAT ZEN	0.9 (n.d.–1.8) 23 (14–55) 151 (48–709) 268 (116–372) 105 (12–890) 49 (5–150)	12 50 50 50 15 50	[36]		AFB1: 0.36 DON: 0.055 FB1:0.709 ZEN: 1.5 to 2
*Kutukutu* (fermented maize-based dough)	North (AEZ 1) West (AEZ 3)	AFB1	(≤2.8)	29	[36]	NA	-
Sorghum (Variety Damugari)	North (AEZ 1)	AFB1	75 (0–230)	NI	[18]	AFB1: 20 µg/kg EU	AFB1: 11.5
Sorghum (Variety Djigari)	North (AEZ 1)	AFB1	45 (0–145)	NI	[18]	AFB1: 20 µg/kg EU	AFB1: 7.25
Sorghum beer (*Bil-bil*)	North (AEZ 1)	DON FB1	450 (140–730) 150 (0–230)	70 55	[18]	DON: 750 µg/kg **	DON: 0.73 to 0.97 FB1: 0.23
Sorghum beer (*Kpata*)	North (AEZ 1)	DON FB1	520 (0–680) 210 (0.5–340)	37 50	[18]		DON: 0.68 FB1: 0.34
Rice	North (AEZ 1)	OTA	(0.2–0.3)	NI	[34]	1250 µg/kg OTA (unprocessed cereals): 5 µg/kg (Codex) **	0.06
Cassava fresh	Littoral (AEZ 4)	OTA	(0.04–0.1)	NI	[34]	5 µg/kg (Codex) **	0.02
Cassava dry	Littoral (AEZ 4)	ZEN	7.6 (NI)	NI	[34]	EU (100 µg/kg)	0.08
Cassava products	North (AEZ 1) West (AEZ 3)	AFB1 Penicillic acid	NI (6–194) NI (25–184)	41 10	[31]	20	AFB1: 9.7
Stored cassava chips	South (AEZ 5)	AFs	(5.2–15)	NI	[40]	20 -	AFs: 0.75 to 3.75
Miscellaneous: Rice Pumpkin seeds (*egusi*) Fermented cassava flakes (*gari*) Fermented cassava flour (*nkum nkum*)	North west (AEZ 3) South west (AEZ 4) Littoral (AEZ 4) Center (AEZ 5)	FB1 ZEN DON AFs	ND 67 (NI) 25 (13–35) 0.3 (NI)	0 1 3 1	[17]	DON: 750 µg/kg **	0.05
Peanuts	Littoral (AEZ 4)	AFs	(14.3–14.6)	NI	[34]	15 µg/kg EAC limits 4 µg/kg **	0.97 to 3.64
Peanuts meal	West (AEZ 3) North west (AEZ 3) Center (AEZ 5) Littoral (AEZ 4)	AFs	161 (39–950)	41	[33]	AFB1: 5 µg/kg AFs: 10 µg/kg **	95 to 190
Groundnuts	West (AEZ 3) Center (AEZ 5)	AFB1	47 (210)	34	[36]	AFs: 10 µg/kg **	NA
Beans	North (AEZ 1)	AFs	(0.2–0.6)	NI	[34]	20 µg/kg US FDA	0.03
	Center (AEZ 5)	FB1 ZEN DON AFs	727 (28–1351) 48 (27–157) 25 (13–35) 2.4 (0.2–6.2)	3 5 7 5	[17]	1000 µg/kg 100 µg/kg 100 µg/kg 20 µg/kg	FB1:1.35 ZEN: 1.57 DON: 0.35 AFs: 0.31
Soybeans	Center (AEZ 5)	FB1 ZEN DON AFs	195 (25–365) 0 110 (13–207) 2.1 (0.2–3.9)	2 0 2 2	[17]	1000 µg/kg 100 µg/kg 100 µg/kg 20 µg/kg	FB1: 0.365 DON: 2.07 AFs: 0.19
Cocoa beans and derived products (roasted cocoa, nibs, butter, cocoa powder, chocolate spread)	South west (AEZ 4)	OTA	11.52 (5.3–21)	NI	[22]	0.5–15 µg/kg EU	1.4 to 42
Green coffee beans	Center (AEZ 5)	OTA	1.7 (1–2.5)	4	[21]	0.5–15 µg/kg EU	0.16 to 5
Arabica coffee	Center (AEZ 5)	OTA	(0.12–124)	NI	[23]	0.5–15 µg/kg EU	8.26 to 248
	West (AEZ 3)	OTA	(0.3–4.9)	33	[23]		0.32 to 9.8
	South west (AEZ 4)	OTA	(0.12–124)	NI	[41]		
Robusta coffee	South west (AEZ 4)	OTA	(0.6–18)	36	[41]	0.5–15 µg/kg EU	1.2 to 36
Black pepper	Center (AEZ 5)	OTA	1.5 (1.2–1.9)	2	[35]	OT: 15 µg/kg	0.12
White pepper	Center (AEZ 5)	OTA	3.3 (1.8–4.9)	8	[35]		0.32
Eggs	Center (AEZ 5)	AFs	0.82 (NI)	28	[42]	NA	NA
Cow milk	Center (AEZ 5)	AFM1	NI (0.006–0.53)	10	[42]	0.5 µg/kg **	1.06
*Irvingia gabonensis* (African mango)	Center (AEZ 5) Littoral (AEZ 4) West (AEZ 3)	AFs Fs ZEN	3.54 (NI) 0.00 (NI) 4.61 (NI)	3 0 0	[43]	AFB1 (spices): 5 µg/kg AFs (spices): 10 µg/kg **	NA
*Aframomum melegueta* (graines du paradis)	Center (AEZ 5) Littoral (AEZ 4) West (AEZ 3)	AFs Fs ZEN	0.32 (NI) 9.30 (NI) 20.24 (NI)	0 0 0	[43]		NA
*Afrostyrax lepidophyllus* (bush onion)	Center (AEZ 5) Littoral (AEZ 4) West (AEZ 3)	AFs Fs ZEN	2.5 (NI) 28.23 (NI) 110.89 (NI)	2 0 2	[43]		NA
*Ricinodendron heudelotii* (Njansang)	Center (AEZ 5) Littoral (AEZ 4) West (AEZ 3)	AFs Fs ZEN	0.63 (NI) 78.62 (NI) 7.84 (NI)	0 0 0	[43]		NA
*Xylopia aethiopica* (guinea pepper)	Center (AEZ 5) Littoral (AEZ 4) West (AEZ 3)	AFs Fs ZEN	1.2 (NI) 891.97 (NI) 219.47 (NI)	2 15 30	[43]	AFB1 (spices): 5 µg/kg AFs (spices): 10 µg/kg **	NA
*Monodora myristica* (African nutmeg)	Center (AEZ 5) Littoral (AEZ 4) West (AEZ 3)	AFs Fs ZEN	0.7 (NI) 6.52 (NI) 110.89 (NI)	0 0 20	[43]		NA
*Tetrapleura tetraptera* (Gum tree)	Center (AEZ 5) Littoral (AEZ 4) West (AEZ 3)	AFs Fs ZEN	0.9 (NI) 437.08 (NI) 52.56 (NI)	0 2 0	[43]		NA
Vegetable oil (cottonseed oil)	Littoral (AEZ 4) North (AEZ 1)	Sterigmatocystin	0.2 0.7	NI	[34]		NA
Tilapia	Center (AEZ 5) West (AEZ 3)	AFs AFB1	4.48 (0.26–8.64) 2.23 (0.13–4.32)	34	[20]	20	AFs: 0.43
Kanga	Center (AEZ 5) West (AEZ 3)	AFs AFB1	5.84 (0.21–11.46) 2.82 (0.10–5.59)	34	[20]	20	AFs: 0.57
Carp	Center (AEZ 5) West (AEZ 3)	AFs AFB1	11.93 (6.56–17.35) 5.87 (3.17–8.67)	34	[20]	20	AFs: 0.86
Catfish	Center (AEZ 5) West (AEZ 3)	AFs AFB1	17.72 (3.62–31.38) 8.81 (1.81–15.69)	34	[20]	20	AFs: 1.56
Poultry feeds		FB1 DON ZEN AFB1	468 (16–1930) 164 (19–495) 4.7 (0.3–28) 40		[32]	1000 µg/kg 100 µg/kg 100 µg/kg 20 µg/kg	FB1: 1.93 DON: 4.95 ZEN: 0.28

QL, quantifiable level; NI, not indicated; NA, not available; ND, not detected. ** Commission Regulation (EC) no 1881/2006 of 19 December 2006 setting maximum levels for certain contaminants in foodstuffs. EAC, East African Community limits; US FDA, US Food and Drug Administration.

When considering AEZ, all the samples were mostly positive with aflatoxins (AFs) and Fumonisins in the western highlands (AEZ 3) than other AEZs, these are characterized by hot climate and high relative humidity. The composite samples with highest ZEA concentrations were collected in Douala: maize (wet season: 7.6 µg/kg; dry season: 97.0 µg/kg) and cassava having undergone a drying process prior to being prepared for consumption (dry season: 7.6 µg/kg).

### 4.2. Spices

Spices are nonleafy parts of the plant that are used in small amounts for food flavoring; they are considered as an essential part of the human diet. Spices are produced mainly in developing countries with tropical climate, where high temperature and humidity facilitate fungal growth and the occurrence of mycotoxins. Few authors reported the contamination of spices in Cameroon. The presence of OTA was detected in 10% of black pepper samples (1.2–1.9 µg/kg) and 40% of white pepper samples (1.8–4.9 µg/kg) collected from Yaoundé markets [39]. Some authors worked on the prevalence and concentrations of AFs, FBs, and ZEN on seven edible nontimber forest products used as spices (*Irvingia gabonensis, Aframomum melegueta, Afrostyrax lepidophyllus, Ricinodendron heudelotii, Xylopia aethiopica, Monodora myristica,* and *Tetrapleura tetraptera*) [43]. Contamination of 5.56% of samples of *A. lepidophyllus*, 8.33% of samples of *X. aethiopica*, and 11.11% of samples of *I. gabonensis* was above the regulatory limit (10 pbb). *Ricinodendron heudelotii*, *M. myristica,* and *T. tetraptera* had AF levels below the regulatory limit. When considering AEZs, all the commodities and AEZs were positive for AF. Occurrence of AFs in *A. melegueta* and *M. myristica* was lower in the three AEZs. Over 84% of the samples had detectable levels of AFs, among which only 5.71% were above the regulatory limit. The majority of commodity samples (94.28%) were safe for human consumption. Fumonisin was not detected in *I. gabonensis* and *M. myristica* samples. The level of FBs was significantly higher in samples of *X. aethiopica* (891.97 µg/kg) followed by *T. tetraptera* (437.08 µg/kg). All samples of *X. aethiopica* had FB content above 100 µg/kg. Fumonisin content was higher in *X. aethiopica* and *T. tetraptera* samples irrespective of AEZs. Among 210 samples collected from the three AEZs, 53% had detectable levels of FBs and only 5% were above the regulatory limit (1000 µg/kg) while 47% were below the limit of detection. The level of ZEN in some commodities was above it. This was the case for *A. lepidophyllus* (5.55%), *M. myristica* (66.67%), and *X. aethiopica* (100%). Zearalenone content was higher in *X. aethiopica* and *M. myristica* samples irrespective of AEZs; however, the level of ZEN in each commodity did not vary significantly between AEZs. Overall, 92% of samples was positive for ZEN among which 21% was above the legal limit (100 µg/kg), despite the regulatory limit (100 µg/kg) being high compared to some other mycotoxins.

### 4.3. Cassava and Derived Products

The Cassava root contributes to food security, income, and employment opportunities in rural areas of sub-Saharan Africa. In Cameroon, cassava roots are processed into fufu, gari, tapioca, starch, and flour. Fresh cassava and cassava products are perceived as potential sources of mycotoxins. Some authors assessed the AFs content in cassava chips, a cassava-derived product (obtained after fermentation and drying) which is widely consumed locally [40]. Aflatoxins and penicillic acid were found at a concentration of 194 µg/kg and 184 µg/kg, respectively, in cassava products (flakes and chips) [31]. The presence of OTA was detected in fresh cassava and ZEN in dried cassava in the center and north of Cameroon at concentrations below the EU limits [34]. The Sudano Sahelian zone (AEZ 1) and the western highlands (AEZ 3) exhibit the highest concentrations for AFB1 and penicillic acid.

### 4.4. Cocoa and Coffee

Cocoa and coffee are food commodities with a high impact on the economy of many producing countries such as Cameroon. Cocoa is an important ingredient in a number of food items, such as cakes and sweets, and coffee is a primary export product in world trade. Due to their high level of consumption, stringent standards have been defined for cocoa and coffee, with strict control in the international markets. In their study on the effect of postharvest treatment on the final OTA content in cocoa beans and their derivatives (roasted cocoa, nibs, butter, cocoa powder, and chocolate spread) [22,41], authors observed that pod damage and late pod opening were aggravating factors for OTA contamination. Fermented dried cocoa from intact pods presented an OTA content below those from poor-quality pods (intentionally or naturally damaged) which showed contents up to 76 µg/kg [41]. Among the 104 samples that they tested, only a few presented an average OTA content above the 5 µg/kg international limit for roasted coffee. Similar observations were made by other authors [42]. The local Arabica coffee brand samples were all below the limit, and only a few from the Robusta coffee were above the limit. Studies conducted by Romani et al. [21] reported an OTA contamination of green coffee beans from Cameroon.

### 4.5. Peanuts, Beans, and Soybeans

Peanut (*Arachis hypogaea L*.) is the most important cultivated grain legume crop and covers an estimated 120,000 hectares annually. Soybeans are today one of the main and important sources of proteins (40–42%) and vegetable oil (18–22%) used in human food [31]. Soybean ranks second in legumes cultivated after peanuts; however, as with many other crops [31,32], its yield is limited in sub-Saharan Africa by several factors, including poor cultivation practices. Beans are the second most important grain crop next to maize in terms of production and consumption in Cameroon, especially in the western highlands. An evaluation of the mycotoxin content in peanuts, soybean, and beans reveals the presence of Aflatoxins (AFs), Fumonisins, ZEN, and DON [17]. OTA was also detected in 13 out of 90 peanut samples [31]. The same authors did not find ZEN in soybeans whereas detected an average content of 46.7 µg/kg in beans. They also detected FB1 in 18 of the 35 peanuts samples and in all 10 soybeans samples at a mean amount of 5 and 49 µg/kg, respectively [32]. Western highlands is found to be the agroecological zone with high concentrations of mycotoxins.

### 4.6. Milk, Eggs, and Fish

The contamination of eggs and cow milk by aflatoxins was reported by Tchana et al. [42]. In lactating animals, the conversion rate of AFB1 to its metabolite AFM1 ranges between 0.5% and 6%. AFM1 is not destroyed during pasteurization and heating process and is found in milk and milk products obtained from livestock exposed to AFB1-contaminated feed [44]. Few authors reported the contamination of fish by aflatoxins. Samples of fish tissue were collected from different farming sites named *Mfou* and *Batié* (center and west Cameroon) [20]. Both AFs and AFB1 were detected in the samples at levels that significantly (*p* < 0.05) change with fish species. Catfish was the most contaminated fish, with AFB1 ranging 1.81–15.69 µg/kg and AF ranging 3.62–31.38 µg/kg. The lowest AFs (0.21 µg/kg) and AFB1 (0.10 µg/kg) levels were observed in kanga. Within each species, a significant variation (*p* < 0.05) in AFs and AFB1 levels was found: 40 times in the case of tilapia, 50 times in the case of kanga, 8.6 times in the case of catfish, and 2.16 times in the case of carp.

### 4.7. Feeds

Cow and beef meat, poultry meat, pork meat, fish, eggs, and milk are among the most frequently consumed foods of animal origin in Cameroon [45]. A concentration of AFs in the range of 39–950 µg/kg was reported in peanuts meal destined for poultry feeds [33]. The levels of mycotoxins in poultry feeds (generally made up to 80% of maize and sorghum) in Cameroon were assessed by Abia et al. [46]. They collected 63 poultry feeds samples from 35 farms situated in areas at high intensity of animal breeding: western highlands (Bamenda, Bafoussam, and Bangante) and southern zone (Yaoundé). All poultry feeds were contaminated by FBs (B1, B2, B3), DON, ZEN, and 20 other mycotoxins that are not yet regulated. With regard to fish farming, in a recent study, they revealed the situation in four regions of Cameroon [47]. Feeding practices are characterized by the use of locally formulated powdered feeds (31.7%), animal manure, chicken droppings (20.5%), and pig dung (18.7%). Feed for fish farming is mainly composed of local cotton seeds, groundnut flour, maize, fish flour, and animal manure.

Hazard index is a proportion number indicating the level of contamination compared to the maximum level; this help to appreciate the possibility of risks. Table 3 provides an overview of dietary exposure to mycotoxins through the consumption of contaminated foods in Cameroon, as well as the expected intake compared to the health guidance available.

## 5. Future Challenges in Cameroon: The Penja Pepper

In the present review, data from white and black pepper suggest a high level of Ochratoxin in the product samples in the center region [35]. No information was provided on the origin of these pepper, whether samples came from Penja pepper or from a location not included in the Penja geographical indications. Since drying, storage, packaging, and transport are critical steps for mycotoxins production, a guide for good sanitary and phytosanitary practices has been produced to empower stakeholders in the pepper value chain. In the hazard analysis, the presence of fungi and mycotoxins has been identified as a point of attention, with a risk characterization score of 8 (gravity of effect 2 and probability of occurrence 2).

Penja pepper is among the best pepper in the world and is much appreciated by top chiefs. It is the first product to obtain a protected geographical indication (PGI) label in sub-Saharan Africa. To be recognized as Penja pepper, one needs to cultivate it in a well-described geographical zone that goes from the Moungo division (Manjo; Mbanga, Nlohé, Loum) to the south-west region (Tombel, Koupé manengouba). The characteristics of the soil and the microclimate of this geographical production area, together with its organoleptic qualities, make Penja pepper an exceptional product. The sector currently has around 450 listed stakeholders, around 20% of which are women, located in five production areas. The identified producers cultivate approximately 420 hectares of pepper. Europe is the main destination market for exports of pepper from Cameroon. Penja pepper may no longer have access to EU markets because of mycotoxins, which is one of the main causes of nonconformities to EU Sanitary and Phyto-Sanitary (SPS) regulations. The review and analysis of notifications of interceptions by the European authorities demonstrate that exports from Cameroon are still facing serious issues to meet SPS constraints. A report of the EU RASFF portal indicates that Cameroon export received 70, 69, and 20 notifications in 2015, 2016, and 2017, respectively, due to the presence of pests and other issues [48]. Moreover, new changes to European regulations on plant health have accentuated the requirements calling for improved measures (a) to manage phytosanitary risks upstream of the supply chain, (b) for the inspection system, and (c) for phytosanitary certification. Sanitary and quality issues are (i) noncompliance with good practices when using pesticides; (ii) no pest control products are currently registered in Cameroon to combat diseases and insects affecting pepper; (iii) noncompliance with good practices during harvesting (Figure 2); (iv) fungus contamination, high humidity, and mycotoxins productions (Figure 3); (v) the risk of contamination by mycotoxins during treatment and storage (Figure 4); (vi) noncompliance with the accepted moisture levels in certain finished products; and (vii) difficulties in the choice of specific packaging for each type of pepper (white, black, and red) (Figure 5).

The ongoing project (STDF project PG/593) funded by the Standards Trade and Development Facility and implemented by COLEACP aims at ensuring that Penja pepper is produced and processed in the best SPS conditions. This will bring the current specifications in line with international SPS regulations, adopting good agricultural practices (GAPs), good phytosanitary practices (GPPs), good hygiene practices (GHPs), and good manufacturing practices (GMPs) based on the Hazard Analysis and Critical Control Point (HACCP) system.

## 6. Discussion

Data provided in Table 2 indicate a general contamination of all food chains. Not only cereals (maize, rice, and sorghum), tubers (cassava), vegetables and fruits (African mango), beans, soybeans, spices and aromatic herbs regularly used in the kitchen (*Aframomum melegueta,* onion sauvage, Njansang, Noix de muscade, quatre cotés, and white pepper), nuts (groundnuts and peanuts), coffee and cocoa, oils, food from animal origins (eggs, cow milk), and fish (tilapia, kanga, carp, and catfish) but also ready-to-eat meals based on maize (e.g., fufu) are contaminated.

Fumonisin B1 had been detected in a number of food items (maize, maize-based dishes, beans, traditional sorghum beer, spices, and poultry feeds); the concentration varied enormously within and between food groups. Levels up to 26,000 µg/kg have been found in maize [19,38]. OTA is the most commonly found OT in foods—both raw and processed—and is the most dangerous in terms of toxicity. Ochratoxin contamination has been found in dried foods, including nuts, beans, fruits, and fish. It can infect wheat and barley crops, and it has also been found in poultry and pork meat [5]. OTA is present in rice, cocoa beans, coffee (Arabica and Robusta), green coffee beans, fresh cassava, and pepper with the highest level found in Arabica coffee (124 µg/kg). Deoxynivalenol, also known as vomitoxin due to its strong emetic effects after consumption, is the most common contaminant of grains and their products. One of the most important physicochemical properties of DON is its ability to withstand high temperatures [49]. Food-producing animals are exposed to mycotoxins through contaminated feeds [50]. Since foods of animal origin are poorly perceived as at risk of contamination, in some cases, feed is not properly stored and could result in mycotoxin contamination [47]. The most important mycotoxins in feed ingredients in terms of risk to fish and consumers are AFB_1_, DON, nivalenol (NIV), ZEN, OTA, T-2 toxin (T2), FB_1_, moniliformin (MON), enniatins (ENNs), and beauvericin (BEA) [51]. Their hazard index could not be estimated because of an absence of suitable maximum regulated limits. The application of usual operations (cleaning, sorting, and milling) and thermal processes can decrease carry-over of mycotoxins from feed to animals [5]. Based on available evidence, foods of animal origins appear to be less contaminated. However, for example, since aquaculture is expected to continue growing in Cameroon, controlling mycotoxins in fish feedings and fish is important. Fish contamination by mycotoxins influences animal well-being, growth, health, as well as nutritional quality for human consumption [52]. Indeed, in the modern concept of zoonosis, exposure to toxicants through foods of animal origin is a well-established issue [53,54]. Mycotoxins negatively affect food safety, food security, and nutrition in sub-Saharan Africa [55], and the changing climate may increase the burden of mycotoxins in feeds and foods [34]. An increasing collection of evidence (e.g., photos and videos characterizing points of particular attention in food chains) can improve awareness of previously unrecognized/overlooked real-life risk scenarios and allow communities to define their specific risk scenarios, prioritize their risks, properly address available resources, and ultimately make their science-informed choice [55]. With an increase in funding for scientific research and infrastructure, formal risk assessment based on evidence will enforce regulations and standards to control mycotoxins in food. For sure, good agricultural and manufacturing practices from field to consumer [56] are crucial.

Sampling is also very important to provide a representative sample, and many sampling procedures exit based on statistical parameters related to consumer safety [57]. An effective sampling procedure is necessary for developing countries since mycotoxins exposure is high in African countries. In the present review, for all the studies selected, the sample size varies widely. It is noticeable that the sample size significantly reduces variation within a sample. Some studies revealed that the higher the number of samples taken, the higher the probability that the food contains mycotoxins. In fact, sample size plays a crucial role in providing an effective sampling for mycotoxins detection [58]. 

Appropriate and timely harvesting can mitigate mechanical injury and insect infestation, while postharvest practices (e.g., rapid and proper drying, postharvest insect control, proper transportation and packaging, good storage conditions and length, and use of natural and chemical agents) and processing (e.g., sorting, cleaning, milling, fermentation, baking, roasting, flaking, nixtamalization, and extrusion cooking) controlling temperature, relative moisture, and humidity can reduce fungal growth and development of mycotoxins [2,44,59]. Regardless of the fact that molds have the genetic potential to produce a particular mycotoxin, the level and rate of production would partly be influenced by available nutrients [60]. As such, different food substrates may have different effects on aflatoxin production due to differences in nutrient content [61]. Besides correct feed storage to prevent mycotoxins, other good practices in the farm await implementation, e.g., proper use of biocides and preserving agents. Clean livestock feed holds the key to clean milk (e.g., artificial drying in the wet seasons); transportation of grains in wet and closed-packed conditions (lack of aeration) is unsafe [14]. The spread and implementation of cost-effective good practices in the Cameroonian AEZs will make a difference in mycotoxin exposure.

Prevention of mycotoxins in food is a matter of primary prevention (exposure of healthy subjects) but also of secondary prevention (e.g., HIV/AIDS subjects with the compromised immune system) [59]. Given the adverse effects (e.g., immune development) in infants and young children, cereal–legume blends could be substituted with root and tuber-based blends (such as sweet potato) in complementary feeding to reduce aflatoxin ingestion [14] and impair nutrition [62]. Some seminal toxicovigilance systems exist in Africa, which deserve development [63]. On the basis of available data in Cameroon, the EDI of major mycotoxins in food for Aflatoxins was 0.0018–14.2 µg/kgbw/day in maize, 0.027–2.36 µg/kgbw/day in cassava, and 0.023–0.1 µg/kgbw/day in groundnuts and fish. For fumonisins, the tolerable daily intake was 0.02–60.6 µg/kgbw/day in maize and 0.056–0.82 µg/kgbw/day in beans. Based on the estimated distribution of human exposure levels by food, maize and cassava are the major sources of exposure and should be prioritized, followed by beans and spices. The Hazard Index provides an indication of the quality of the products and possible health risks to the consumer. The maximum level is the limit or the maximum quantity of mycotoxin set by countries. These limits are not set for Cameroon; therefore, what should be applied are limits set by the *Codex Alimentarius*. Where Codex Maximum limits do not exist, we used the ones of the European Union since it is one of the main markets for exportation and food security (food waste, loss, and health of animals). Finally, the economic impact on commercial trade should also be assessed and monitored. According to available data, Cameroonian maize products, sorghum, cacao, peanuts, spices, and some locally grown fish will not be able to enter European markets. Mycotoxins represent the group of food chemicals which triggered the most cases of border rejection (489) recorded in the European Union (EU) Rapid Alert System on Food and Feed [48]. Only in 2019, mycotoxin notifications were the second most important with 588 notifications, after pathogenic microorganisms [48]. African exports have been rejected many times at the EU borders because of their noncompliance with EU product standards. A study showed that between 2008 and 2013, an analysis of the EU RASFF platform revealed that out of a total of 20 reasons, mycotoxin contamination is the major reason for border rejection of export products coming from Africa to enter EU markets, i.e., 23% contribution [64]. Most concerned products were, e.g., (i) nuts, nuts products, and seeds; (ii) fruits and vegetables; (iii) fish and aquatic products; (iv) herbs and spices; (v) dietetic foods; (vi) poultry meat; and (vii) cocoa, coffee, and tea. These results comply with a recent study by Alshannaq and Yu [64] who analyzed the RASS system from 2010 to 2019 for AFs notifications from US export in the EU, as well as the findings of this review.

## 7. Conclusions

Very few data exist on mycotoxin contamination in Cameroonian agricultural commodities and food items. From 2000 to 2021, we found only 25 published studies from 14 different authors. The 14 authors are affiliated with different universities among which some are foreign universities. The laboratory of toxicology at the University of Yaoundé, Cameroon, has recently executed some intensive work on the topic, but the amount of data and knowledge remains incomplete to provide a clear view of the national situation. The present study is aimed to contribute a national database of chemicals in foods towards national risk management of residues and contaminants in food. The evidence reported here for mycotoxins is expected (i) to motivate stricter and wider analytical control on the territory, (ii) to boost local research on these natural toxins in terms of food safety and nutrition, but also food security and associated economic impact, (iii) to further motivate the spreading of good practices and HACCP-based approaches to avoid them (at the level of food producers, retailers, food transformers and vendors, and consumers), and (iv) to improve efforts in national risk assessment and regulatory strategy.

## Figures and Tables

**Figure 1 foods-12-01713-f001:**
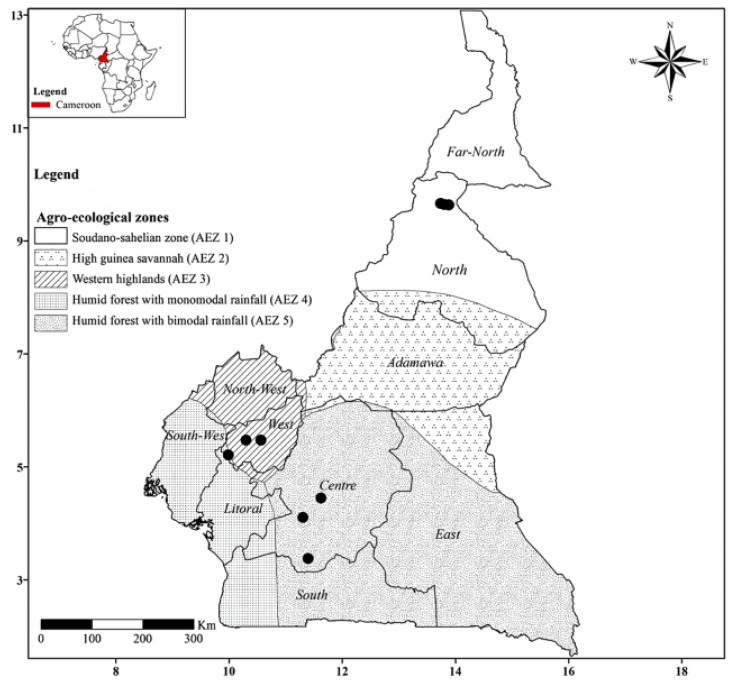
Presentation of agroecological zones of Cameroon.

**Figure 2 foods-12-01713-f002:**
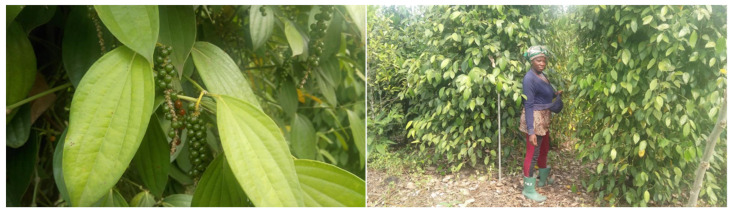
Harvesting pepper without any protective equipment and collecting in clothes (Njombe, Cameroon).

**Figure 3 foods-12-01713-f003:**
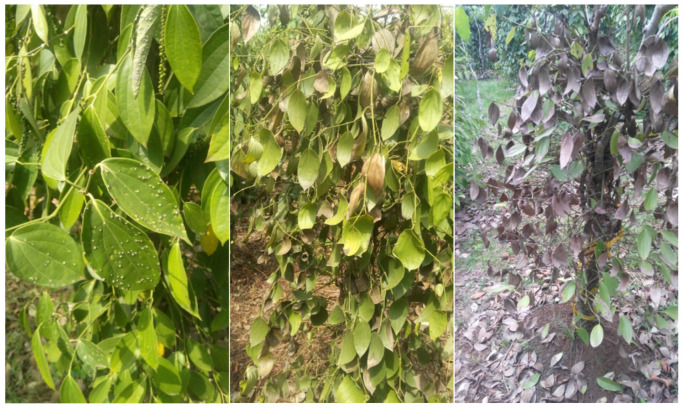
Nematodes facilitate gall diseases and fungi contamination; a defoliated plant after fungi attack (Njombe, Cameroon).

**Figure 4 foods-12-01713-f004:**
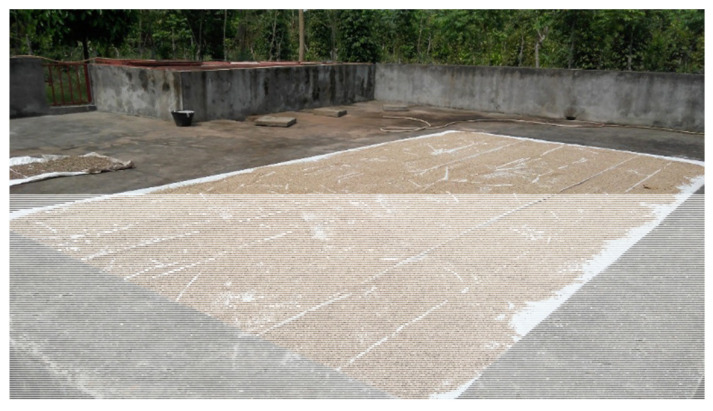
Drying pepper in sunlight (Njombe, Cameroon).

**Figure 5 foods-12-01713-f005:**
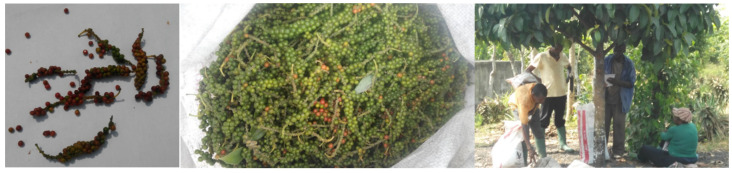
Mature green and red pepper collected in reused plastic bags (Njombe, Cameroon).

**Table 1 foods-12-01713-t001:** Major crops cultivated and animal species reared in each agroecological zone.

Agroecological Zones (AEZs)	Main Crops and Animal Production
Sudano-Sahelian (AEZ 1)	Maize, millet–sorghum, rice, cowpea, soybean, onion, sesame, fruits, cotton, cattle, and small ruminants
High Guinea Savanna (AEZ 2)	Maize, yam, cassava, sweet potatoes, rice, cotton, cattle, pig, small ruminants, and poultry birds
Western Highlands (AEZ 3)	Maize, beans, potatoes, rice, sweet potatoes, vegetables, coffee, pig, poultry, cattle, small ruminants, and fisheries
Mono-modal Humid Forest (AEZ 4)	Banana, plantain, cassava, cocoyam, sweet potatoes, maize, vegetables, cocoa, coffee, oil palm, rubber, fruits, poultry, pig, poultry birds, small, ruminants, and fisheries
Bimodal Humid Forest (AEZ 5)	Plantain, cassava, banana, maize, cocoyam, sweet potatoes, cocoa, oil, palm, rubber, coffee, maize, cocoa, oil palm, fruits, poultry, pig, fisheries, and small ruminants

**Table 3 foods-12-01713-t003:** Dietary exposure to mycotoxins through the consumption of contaminated foods in Cameroon. Expected intake compared to health guidance available.

Foodstuff/Agroecological Zone	Food Consumption (Mean, kg/Person/Day)		Contamination (µg/kg)			Estimated Daily Intake (µg/kg bw/day) (Average bw = 60 kg)						Total Daily Intake (µg/kgbw/day) %TDI = (EDI/TDI) × 100		
	FAO-FBS (2018)	TDS (2008)	Min	Mean	Max	Min		Mean		Max		TDI	%TDI-FBS (Mean)	%TDI-TDS (Mean)
						FBS	TDS	FBS	TDS	FBS	TDS			
Maize	0.14	0.023	AFB1: 0.8	-	20	0.0018	0.003	-	-	0.046	0.007	NE	-	-
			FUM: 50	3684	26	0.12	0.02	8.6	1.4	60.6	9.96	2	430	70
			DON: 100	-	1300	14	2.3	-	-	182	29.9	1	1400	230
			ZEA: 27	69	334	0.063	0.01	0.16	0.026	0.78	0.13	0.5	32	5.2
Rice	0.11123	0.201	OTA: 0.2	-	0.3	0.0003	0.0007	-	-	0.0005	0.001	0.014	3.6	7.1
Cassava	0.265	0.73	AFB1: 6	-	194	0.027	0.073	-	-	0.86	2.36	NE	-	-
Beans	0.0368	0.012	AFL: 0.2	2.4	6.2	0.0001	0.00004	14.72	0.0005	0.004	0.0012	NE	-	-
			FUM: 28	727	1351	0.017	0.0056	0.45	0.15	0.82	0.27	2	22.5	7.5
			ZEA: 27	48	187	0.017	0.005	0.029	0.01	0.11	0.04	0.5	5.8	2
			DON: 13	25	35	0.008	0.0026	0.015	0.005	0.021	0.007	1	1.5	0.5
Peanuts (meal)	NA	NA	AFs: 39	161	950	-	-	-	-	-	-	NE	-	-
Groundnuts	0.0293	0.021	AFB1: 47	-	210	0.023	0.016	-	-	0.1	0.07	NE	-	-
Soybeans	NA	NA	FUM: 25	195	365	-	-	-	-	-	-	2	-	-
			ZEA: 13	110	207	-	-	-	-	-	-	0.5	-	-
			DON: 0.2	2.1	3.9	-	-	-	-	-	-	1	-	-
Coffee	0.0015	0.0018	(Arabica): OTA: 0.12	-	124	0.000003	0.000004	-	-	0.0031	0.004	0.014	22.1	28.6
			(Robusta) OTA: 0.6	-	18	0.000004	0.000018	-	-	0.00045	0.00054	0.014	3.2	3.86
			OTA: 0.04	-	0.1	0.0001	0.001	-	-	0.0004	0.001	0.014	2.85	7.14
Pepper	0	0.0005	(Black) OTA: 1.2	1.5	1.9	0	0.00001	0	0.000013	0	1.6	0.014	0	0.09
			(White) OTA: 1.8	3.3	4.9	0	0.000015	0	0.000028	0	4.08	0.014	0	0.2
Eggs	0.0012	0.0039	AFs: -	0.82	-	-	-					NE		
Cow milk	NA	NA	AFM1: 0.06	-	0.53	-	-	-	-	-	-	NE	-	-
Spices (Njansang)	0.003	0.0012	AFs: -	0.63	-	-	-	0.000032	0.000013	-	-	NE	-	-
			FUM: -	78.62	-	-	-	0.0039	0.0016	-	-	2	0.195	0.08
			ZEA: -	7.84	-	-	-	0.00039	0.00016	-	-	0.5	0.078	0.032
Vegetable oil	0.004	0.0078	Sterigmatocystin: -	0.7	-	-	-	0.000047	0.0001	-	-	NE	-	-
Tilapia	0.0045	0.0014	AFs: 0.26	4.48	8.64	0.00002	0.000006	0.0003	0.0001	0.0006	0.0002	NE	-	-
			AFB1:0.13	2.23	4.32	0.000009	0.000003	0.0002	0.00005	0.0003	0.0001	NE	-	-
Kanga	0.0045	0.00046	AFs: 0.21	5.84	11.46	0.000015	0.000002	0.00044	0.00004	0.0009	0.0009	NE	-	-
			AFB1: 0.1	2.82	5.59	0.0000075	0.0000007	0.0002	0.00002	0.00042	0.00004	NE	-	-
Carp	0.0217	0.00046	AFs: 6.56	11.93	17.35	0.0023	0.000005	0.004	0.000009	0.006	0.00013	NE	-	-
			AFB1: 3.1	5.87	8.67	0.0011	0.000002	0.002	0.000004	0.003	0.00007	NE	-	-
Catfish	0.0045	0.0014	AFs: 3.62	17.72	31.38	0.0003	0.000008	0.001	0.0004	0.0023	0.0007	NE	-	-
			AFB1: 1.81	8.81	15. 69	0.0001	0.000004	0.0007	0.0002	0.0012	0.0003	NE	-	-

NE: Not Established.

## Data Availability

Data are contained within the article.

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
