# Peer review of "Occurrence and Dietary Risk Assessment of Mycotoxins in Most Consumed Foods in Cameroon: Exploring Current Data to Understand Futures Challenges"

_foods, 2023, doi:10.3390/foods12081713_

Round 1

Reviewer 1 Report

The review article “Occurrence and dietary risk assessment of Mycotoxins in most consumed foods in Cameroon: exploring current data to under-3 stand futures challenges” describes the overall status of mycotoxin contamination of different crops in Cameroon. The article provides important information for the readers of the field. However, there are many issues with the manuscript which need to be taken care of:

1.      English editing is required for the manuscript.

2.      There are many typos or unwanted errors throughout the manuscript, which need to be corrected carefully.

E.g. Page 2 line 46 “Ochratoxine”

Page 6 line 214 “Total”

Page 6 line 216 need to connect the para

Page 6 line 220 “ 26.000”? should be 26.0

Page 8 line 300 “ food (). soybean ranks 2nd in legumes cultivated after peanuts, However, as with many”.

And so on throughout the manuscript. Correct carefully!

3.      The numbering of the sections and subsections is confudsing. Eg. 1 is given to many sections, after 1.1 suddenly 3.2 comes.

4.      The abbreviation need to be describes at the first place of its use and then only use abbreviations. Need to check carefully and correct accordingly.

5.      What is JECFA?

6.      On page 15, table 2 caption is different form cation on page 16. Check

7.      Give forms of the abbreviations at the foot note of table.

8.      Remove the unnecessary spaces in the discussion section.

9.      In discussion “Deoxynivalenol, also known as vomitoxin due to his strong emetic effects”, should be “its”.

10.  The case description should be given before the discussion.

11.  Incorporate the references in the case description also.

12.  Reference section is missing.

Author Response

Response to Reviewer 1 Comments

Point 1: English editing is required for the manuscript.

Response 1: Ok

Point 2: There are many typos or unwanted errors throughout the manuscript, which need to be corrected carefully.

E.g. Page 2 line 46 “Ochratoxine”

Page 6 line 214 “Total”

Page 6 line 216 need to connect the para

Page 6 line 220 “ 26.000”? should be 26.0

Page 8 line 300 “ food (). soybean ranks 2nd in legumes cultivated after peanuts, However, as with many”. And so on throughout the manuscript. Correct carefully!

Response 2:  all errors were corrected

Point 3: The numbering of the sections and subsections is confusing. Eg. 1 is given to many sections, after 1.1 suddenly 3.2 comes.

Response 3:  Thank’s I correct the numbering of each sections, now it’s OK

Point 4: The abbreviation need to be describes at the first place of its use and then only use abbreviations. Need to check carefully and correct accordingly.

Response 4 :  now it’s OK

 Point 5 :   What is JECFA?

Response 5 :  JECFA is the Joint FAO/ WHO expert Committee on Food Additives

 Point 6 : On page 15, table 2 caption is different form cation on page 16. Check

Response 6 :  JECFA is the Joint FAO/ WHO expert Committee on Food Additives

 Point 7 : Give forms of the abbreviations at the foot note of table.

Response 7 :  OK, done

  Point 8 :  Remove the unnecessary spaces in the discussion section.

Response 8 :  OK, done

Point 9 :    In discussion “Deoxynivalenol, also known as vomitoxin due to his strong emetic effects”, should be “its”.

Response 9 :  OK, done

Point 10 :  The case description should be given before the discussion.

Response 10 :  OK, I correct in the manuscript

Point 11 : Incorporate the references in the case description also.

Response 11 :  OK, I correct in the manuscript

 Point 12 :  Reference section is missing.

Response 12 :  there is a references section in the Manuscript at the end

Reviewer 2 Report

Studies on mycotoxins, especially in agricultural products in developing countries, are valuable. The study has useful information for researchers and food policymakers in Cameroon and developing countries. The manuscript has some errors that need to be fixed. Please, the authors pay attention to the highlighted words and correct the writing errors. Similarly, comments are sent in the file to be considered. Thanks

Author Response

Response to Reviewer 2 Comments

Point 1:    remove Cameroon and add another keywords

Response 1:  I  removed it   and add another

Point 2: There are many typos or unwanted errors throughout the manuscript, which need to be corrected carefully.

Response 2:  all errors were corrected

Point 3: provide references.

Response 3:   All the references are provided

Point 4: remove risk analysis

Response 4 :  now it’s OK I corrected in the manuscript

 Point 5 :   Use the word local beer

Response 5 :  I correct in the manuscript

 Point 6 : explain Hazard index and the abbreviations at the foot note of table.

Response 6 :  OK, done

Hazard Index is a proportion number indicating the level, of contamination compare to the maximum level; this help to appreciate the possibility of risks.  It was calculated using the formula

Ql max/Maximum limits, Qlmax is the maximum residue level on foods. All the abbreviations were explain on foot note table

Point 7: Units for TDI

Response 7: that is Total daily intake is expressed in µg/kgbw/day
